# Germination Stimulant Activity of Isothiocyanates on *Phelipanche* spp.

**DOI:** 10.3390/plants11050606

**Published:** 2022-02-24

**Authors:** Hinako Miura, Ryota Ochi, Hisashi Nishiwaki, Satoshi Yamauchi, Xiaonan Xie, Hidemitsu Nakamura, Koichi Yoneyama, Kaori Yoneyama

**Affiliations:** 1Graduate School of Agriculture, Ehime University, Matsuyama 790-8566, Japan; h652021a@mails.cc.ehime-u.ac.jp (H.M.); ryoutaochi.org.chem@gmail.com (R.O.); nishiwaki.hisashi.mg@ehime-u.ac.jp (H.N.); yamauchi.satoshi.mm@ehime-u.ac.jp (S.Y.); 2Center for Bioscience Research and Education, Utsunomiya University, Utsunomiya 321-8505, Japan; xie@cc.utsunomiya-u.ac.jp (X.X.); yone2000@sirius.ocn.ne.jp (K.Y.); 3Graduate School of Agriculture and Life Sciences, The University of Tokyo, Tokyo 113-8657, Japan; hidemitsunakamura87@gmail.com; 4Japan Science and Technology, PRESTO, Kawaguchi 332-0012, Japan

**Keywords:** germination stimulant, isothiocyanates, *Phelipanche*, structure–activity relationship, suicidal germination

## Abstract

The root parasitic weed broomrapes, *Phelipanche* spp., cause severe damage to agriculture all over the world. They have a special host-dependent lifecycle and their seeds can germinate only when they receive chemical signals released from host roots. Our previous study demonstrated that 2-phenylethyl isothiocyanate is an active germination stimulant for *P. ramosa* in root exudates of oilseed rape. In the present study, 21 commercially available ITCs were examined for *P. ramosa* seed germination stimulation, and some important structural features of ITCs for exhibiting *P. ramosa* seed germination stimulation have been uncovered. Structural optimization of ITC for germination stimulation resulted in ITCs that are highly active to *P. ramosa*. Interestingly, these ITCs induced germination of *P. aegyptiaca* but not *Orobanche minor* or *Striga hermonthica*. *P. aegyptiaca* seeds collected from mature plants parasitizing different hosts responded to these ITCs with different levels of sensitivity. ITCs have the potential to be used as inducers of suicidal germination of *Phelipanche* seeds.

## 1. Introduction

Broomrapes (*Orobanche* and *Phelipanche* spp.) in the family Orobanchaceae are devastating obligate root parasitic weeds damaging crop production all over the world [1,2]. In general, broomrapes have wide host ranges including legumes, Solanaceae, Asteraceae, and Brassicaceae, etc. The area threatened by broomrapes, as estimated in 1991, is 16 million ha in the Mediterranean and west Asia [3] and is still increasing.

A single root parasitic weed produces up to 100,000 tiny seeds and it is nearly impossible to remove these seeds from infested soils [4]. The seeds of root parasitic weeds can germinate only when they receive host-derived chemical-germination stimulants. This is a sophisticated strategy for survival of the root parasites, since germinated seeds with limited food stock should attach to the host roots within a couple of days, otherwise they will die. Then, only the parasite seeds in the rhizosphere of host plants will germinate by sensing host-derived stimulants [5]. Accordingly, in severely infested soils, one potentially effective control method for root parasitic weeds is to induce seed germination of the parasites in the absence of host plants, termed ‘suicidal germination’ [6]. In addition, suicidal germination contributes to reduce the parasite seed bank.

Among host-derived germination stimulants for root parasitic weeds, strigolactones (SLs) are the most potent stimulants and are distributed widely in the plant kingdom [5,7]. SLs also act as host recognition signals for symbiotic arbuscular mycorrhizal (AM) fungi in the rhizosphere and are a class of plant hormones that regulate plant architecture and development in plants [4,6]. Strigol was the first isolated SL germination stimulant for *Striga* from root exudates of cotton [8]. So far, more than 30 SLs have been characterized from root exudates of various plant species [9,10]. Importantly, plants exude SLs for AM fungi [11,12] and nitrogen-fixing bacteria [13,14] but not for root parasitic weeds [7,10,15,16]. AM fungi supply mineral nutrients including phosphate and nitrogen to hosts and in turn receive photosynthates from the hosts. More than 80% of land plants form symbiotic relationships with AM fungi [17]. Although AM symbiosis is a reasonable strategy for land plants to effectively obtain mineral nutrients in nutrient limited soils, there are several exceptional plant species which do not form AM symbiosis. The Brassicaceae plants including *Brassica* spp., *Arabidopsis* and winter vegetables like cabbage and broccoli are representative non-mycotrophic plants [17]. As SLs function as plant hormones regulating plant architecture and development, even non-mycotrophic plants produce SLs [10]. In general, host plants of AM fungi enhance SL exudation under nutrient starved conditions but non-hosts do not [15].

*Phelipanche ramosa* causes severe damage to oilseed rape (*Brassica napus*) in southern France. Since oilseed rape is a non-host of AM fungi, this plant is expected to exude only small amounts of SLs [15]. Thus, it was suggested that oilseed rape plants may exude non-SL germination stimulants. Indeed, our previous study demonstrated that 2-phenylethyl isothiocyanate (ITC) is an active germination stimulant for *P. ramosa* in root exudates of oilseed rape [18]. ITCs are enzymatically formed from glucosinolates which are specific plant secondary metabolites of *Brassica* species while the roles of ITCs in plants are not fully understood [19].

In the present study, various ITCs including 2-phenylethyl ITC were examined for *P. ramosa* seed-germination stimulation to characterize the important structural features for the activity. Some ITCs were found to be highly active to *P. ramosa* and also to *P. aegyptiaca*, inducing germination at as low as 10^–15^ M.

## 2. Results

### 2.1. Structure–Activity Relationships of ITCs in Germination Stimulation of *Phelipanche* spp.

Germination stimulation activities of 21 commercially available ITCs (Figure 1A) to *P. ramosa* seeds are shown in Figure 2A. Although alkyl ITCs with hexyl or a shorter alkyl group were very weak stimulants, ITCs with a heptyl (C_7_) to dodecyl (C_12_) but not a tetradecyl group showed strong germination stimulation activities. Benzyl ITC was as active as 2-phenylethyl ITC, while phenyl ITC was inactive.

The 3-phenylpropyl, 4-phenylbutyl, 5-phenylpentyl, and 6-phenylhexyl ITCs were prepared and examined for their germination stimulation activity on *P. ramosa* (Figure 1B). Germination stimulation activity of these phenylalkyl ITCs was increased with an increase of the alkyl chain length, and reached a maximum with 4-phenylbutyl ITC and 5-phenylpentyl ITC, being as active as GR24, and dropped dramatically with 6-phenylhexyl ITC (Figure 2B). The germination stimulation activity of 3-phenylpropyl, 4-phenylbutyl, and 5-phenylpentyl ITCs were compared with GR24 at ≤10^−9^ M, and 5-phenylbutyl ITC appeared to be the most active eliciting about 20% germination even at 10^−15^ M (Figure 3).

In *P. aegyptiaca*, structural requirements of ITCs for germination stimulation were similar to those in *P. ramosa* but not the same (Figure 4A). For example, hexyl ITC induced less than 10% germination of *P. ramosa* (Figure 2A) but elicited about 40% germination of *P. aegyptiaca* (Figure 4A). In contrast, dodecyl ITC induced about 70% germination of *P. ramosa* (Figure 2A) but was almost inactive to *P. aegyptiaca* (Figure 4A). Although benzyl ITC was as active as 2-phenylethyl ITC to *P. ramosa* (Figure 2A), it was less active on *P. aegyptiaca* (Figure 4A). Among phenylalkyl ITCs (Figure 1B), 6-phenylhexyl ITC was inactive and both 4-phenylbutyl ITC and 5-phenylpentyl ITC showed high stimulation activities (Figure 4B) as in the case of *P. ramosa* (Figure 2B).

None of the ITCs tested elicited germination of *Orobanche minor* or *Striga hermonthica* seeds (data not shown).

### 2.2. Host Preference

The seeds of *P. aegyptiaca* collected from mature plants parasitizing different hosts, cabbage, tomato, and chickpea, responded to ITCs (10^−9^ M) with different levels of sensitivity, with those from cabbage hosts most sensitive to ITCs (Figure 5). The seeds from chickpea hosts were moderately sensitive and those from tomato hosts were least sensitive to ITCs. It should be noted that the seeds of *P. aegyptiaca* collected from tomato hosts responded to ITC but not to GR24 at 10^−9^ M.

### 2.3. Residual Activity and Effects on Germination and Growth of Cabbage

The ITCs 4-Phenylbutyl and 5-phenylpentyl maintained high activity (about 50% that of day 0) after an incubation for 2 weeks in vermiculite (Appendix A). These ITCs did not negatively influence germination or growth of the host plant cabbage (Appendix A).

## 3. Discussion

Germination stimulation activity of 21 commercially available ITCs indicate that ITCs need an appropriate lipophilicity for high germination stimulation activity and the relatively bulky benzene ring should be separated from the NCS group by at least one carbon (Figure 2). In our previous study, 4-pentenyl ITC was as active as 2-phenylethyl ITC in inducing seed germination of *P. ramosa* [18]. However, in the present study, this ITC was almost inactive (Figure 2). In addition, *P. aegyptiaca* seeds collected from tomato hosts hardly reacted to GR24 in the present study but were rather sensitive to GR24 in the previous study. Such a discrepancy may be due to the different sample application methods in the germination assays; in the previous study, each chemical was applied as an aqueous solution containing 0.1% acetone to the conditioned parasite seeds, whereas in the present study, an aliquot of sample acetone solution was applied to a petri dish lined with a filter paper, water was added to the petri dish after evaporation of the solvent, and then glass fiber disks carrying the conditioned seeds were placed on the filter paper [20]. These differences in the sensitivity to stimulants in the two assay methods, however, may not significantly affect structural requirements in ITCs for germination stimulation.

Alkyl ITCs with an alkyl chain length of C_7_ to C_10_ and phenylalkyl ITCs with an alkyl chain length of C_2_ to C_5_ are highly active germination stimulants for both *P. ramosa* and *P. aegyptiaca*. Therefore, structural requirements of ITCs for *P. ramosa* seed germination are similar to those for *P. aegyptiaca*, but there are clear differences between the two parasitic weed species. For example, at 10^−6^ M hexyl ITC was almost inactive to *P. ramosa* but induced about 20–40% germination in *P. aegyptiaca*. In contrast, dodecyl ITC at 10^–6^ M elicited more than 60% germination in *P. ramosa* but less than 10% germination in *P. aegyptiaca*.

In the case of phenylalkyl ITCs, the benzene ring needed to be separated from the ITC group at least by one carbon atom for high germination stimulation activity. This may suggest that an insertion of an alkyl chain of a proper length between a cycloalkane ring and the ITC group would also enhance the germination stimulation activity (Figure 2A,B and Figure 4A,B). Alternatively, introduction of proper substituents on the benzene ring in the phenylalkyl ITCs may afford highly active stimulants.

One of the promising strategies for mitigating serious damage caused by the root parasitic weeds is to induce suicidal germination in soils [6,19,21]. Sphynolactone-7 (SPL7) was first reported as a femtomolar-range suicide germination stimulant for *S. hermonthica*, which was selected by chemical screening and modified for further activity enhancement [22]. SPL7 possesses methylfuranone but lacks the enol ether bridge, resulting in its enhanced stability. Indeed, in *Striga*-contaminated soil, SPL7 treatment at a concentration of 10^–10^ M a week before planting host maize significantly reduced the emergence of *Striga*. By contrast, GR 24 required 10^−8^ M to achieve similar effects.

In the case of *Striga* seed, ethylene also stimulated germination [6]. Ethylene diffuses widely in the soil, and then ethylene fumigation in *Striga* infested fields has lead to a 90% reduction in viable *Striga* seeds in USA.

Results obtained in in vitro assays do not always support effectiveness in the field, and it is important to examine if the candidate chemicals would work in the fields. GR24, one of the most stable SL analogs, decomposed rather rapidly under field conditions [23,24]. As shown in Appendix A, phenylalkyl ITCs seem to be more stable than SLs in soil, although GR24 was not included in the experiment. In particular, 4-phenylbutyl ITC and 5-phenylpentyl ITC maintained germination stimulation activity about half that of day 0 even after a 14-day incubation.

Methyl ITC generators like dazomet are effective against nematodes and are used widely [25], and in general ITCs are known to exhibit antimicrobial activity [26]. Phenylalkyl ITCs, in particular, benzyl ITC have been reported to be a more active antibiotic than alkyl or alkenyl ITCs including allyl ITC which is known to possess antimicrobial activity against various microorganisms [27,28,29]. Since benzyl ITC has been shown to be a more active antibiotic than 2-phenylethyl ITC, phenylalkyl ITCs with longer alkyl chains would be weak antimicrobial agents. Therefore, 4-phenylbutyl and 5-phenylpentyl ITCs may be used as suicidal germination inducers for *Phelipanche* spp. but they are only weak antimicrobial agents. There have been several attempts to treat parasitic weed-infested fields with suicidal germination inducers prior to planting host crops [30,31]. In these cases, phytotoxicity of residual germination inducers may cause adverse effects on host crops. The ITCs 4-phenylbutyl and 5-phenylpentyl appeared to be safe for at least one host crop, cabbage (Appendix A).

None of the ITCs examined in this study induced seed germination of *O. minor* or *S. hermonthica* (data not shown). These results demonstrate that ITCs are important germination stimulants for *Phelipanche* spp. which have developed a special seed germination strategy to parasitize *Brassica* spp., non-host plants of AM fungi which exude only small amounts of SLs [15]. Interestingly, there were differences in the sensitivity to ITCs among the seeds of *P. aegyptiaca* parasitizing different host crops (Figure 5); seeds collected from cabbage hosts were highly sensitive to ITCs as compared to the seeds from tomato and chickpea hosts. These results indicate that ITCs would be germination stimulants for *P. aegyptiaca* in the rhizosphere of cabbage. Although tomato and chickpea do not release ITCs and their major germination stimulants are SLs, *P. aegyptiaca* parasitizing these hosts still retains a moderate sensitivity to ITCs. In the case of *P. ramosa*, the seeds collected from different hosts, tobacco and oilseed rape, showed different sensitivities to GR24 [32].

The receptor of SLs in higher plants is the α/β-hydrolase D14, while SL receptor of root parasitic plant is the homolog of D14, namely KARRIKIN INSENSITIVE2/HYPOSENSITIVE TO LIGHT (KAI2/HTL) [33]. Intriguingly, *S. hermonthica* has eleven *KAI2/HTL* genes and six of them are developed to be highly sensitive to SLs [34]. The expansion of *KAI2/HTL* genes is also observed in *Phelipanche*; five orthologs were identified from both *P. aegyptiaca* [35] and *P. ramosa* [36]. In *P. ramosa*, PrKAI2d3 is likely to be involved in seed germination elicited by both SLs and ITC [36]. PrKAI2d3 has the ability to enzymatically interact with not only SLs but also ITCs, while involvement of other PrKAI2d needs to be clarified. Synthetic SL GR24 was 10,000-fold more active than the ITCs in both germination stimulation and interaction with the receptor PrKI2d3. The high germination stimulation activities of ITCs observed in our experiments suggest that these ITCs with an appropriate lipophilicity would permeate into lipid-rich parasite seeds more easily than do SLs [37]. Further structural optimization of ITCs, for example, introduction of substituent(s) onto the benzene ring of 4-phenylbutyl ITC and 5-phenylpentyl ITC (Figure 1B) may afford more active ITCs with longer residual activities.

Recently, non-SL germination stimulants for *Orobanche minor* have been isolated from *Streptomyces albus* J1074 [38] and tryptophan derivatives have been shown to induce *O. minor* seed germination [39]. Although these compounds required high concentrations to induce germination, structural modifications may enhance their activities. It remains unclear if these compounds induce germination of other root parasitic weeds. It is interesting that each root parasitic plant species appears to have flexibly evolved to become sensitive to specific chemicals in their specific environments.

## 4. Materials and Methods

### 4.1. Plant Material

*P. ramosa* seeds were collected from mature flowering spikes that were parasites of oilseed rape grown at Saint-Martin-de Fraigneau, France. *P. aegyptiaca* seeds were collected in tomato, cabbage, and chickpea crops at Golan Heights, Western Negev, and Western Galilee, respectively, in Israel. *O. minor* was collected from a red clover field at Utsunomiya in Japan and *S. hermonthica* from a maize field near Wad Medani in Sudan. Seeds of cabbage were obtained from a local supplier.

### 4.2. Chemicals

A total of 21 commercially available ITCs were obtained from Tokyo Chemical Industry Co. Ltd. (Tokyo, Japan). GR24 was kindly supplied by Prof. Kohki Akiyama (Osaka Prefecture University, Osaka, Japan). The other analytical grade chemicals were obtained from Kanto Chemical Co. Ltd. (Tokyo, Japan).

### 4.3. Synthesis

The ITCs 3-Phenylpropyl-, 4-phenylbutyl-, 5-phenylpentyl- and 6-phenylhexyl were prepared through the decomposition of dithiocarbamic acid salts generated in situ by the treatment of the corresponding amines with carbon disulfide and triethylamine [40,41] (Appendix A).

### 4.4. Germination Assay

The germination assay was conducted as reported previously [20]. ITCs were dissolved and diluted by using acetone. An aliquot (<10 µL) of the respective ITC solution was added to each petri dish (i.d. 5 cm) lined with a filter paper. The solvent was allowed to evaporate before conditioned seeds were placed on the filter paper and treated with distilled water (0.65 mL). The seeds of *P. ramosa*, *P. aegyptiaca*, *O. minor*, and *S. hermonthica* seeds were incubated at their optimal temperature, 23, 25, 23, and 30 °C, respectively.

### 4.5. Residual Activity

ITCs dissolved in water (100 mL) were applied onto pots (i.d. 10 cm, 10 cm deep) filled with vermiculite (300 g). Pots were incubated for 2, 7, and 14 days without any host plants under 150 µm m^−2^ s^−1^ (16 h light and 5 h dark) at room temperature. After incubation, the pot was washed with 200 mL of water, the drain water was collected, and extracted with 100 mL of ethyl acetate three times. The ethyl acetate solutions were combined, dried over anhydrous Na_2_SO_4_, and concentrated in vacuo. The ethyl acetate extracts were examined for germination stimulation of *P. aegyptiaca* as in Section 4.4.

### 4.6. Effects of ITCs on Germination and Growth of Cabbage

Cabbage seeds were incubated with 4-phenylbutyl and 5-phenylpentyl ITC (10^−^^8^–10^−^^6^ M) at 20 °C for 5 days and the germination rate and shoot and root lengths were determined.

### 4.7. Statistical Analysis

All experimental results were subjected to ANOVA utilizing JMP software, version 5.0 (SAS Institute Inc., Cary, NC, USA).

## 5. Conclusions

We found highly active germination stimulant ITCs to *Phelipanche* spp., which induce germination as low as at 1 femtomolar. They seem to be relatively stable in soils and do not negatively affect germination or growth of the cabbage host plant. These results suggest that these ITCs would be good lead compounds for suicidal germination inducers to control *Phelipanche* spp. in the field. The seeds of *P. aegyptiaca* collected from mature plants parasitizing cabbage, tomato, and chickpea were more sensitive to these ITCs than the synthetic SL GR24. Hence, ITCs have the potential to be used as inducers of suicidal germination of *Phelipanche* spp.

## Figures and Tables

**Figure 1 plants-11-00606-f001:**
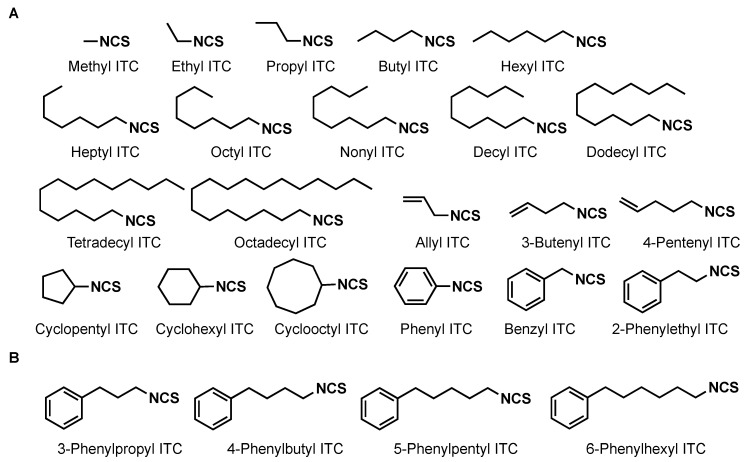
Chemical structures of isothiocyanates used in this study. (**A**) Commercially available products. (**B**) Synthesized in this study.

**Figure 2 plants-11-00606-f002:**
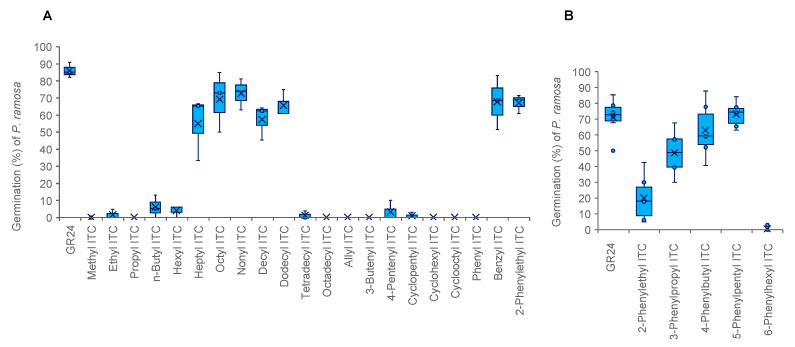
Germination stimulation activities of ITCs on *P. ramosa* seeds. The activities were examined at 10^−6^ M (**A**) and 10^−8^ M (**B**). The box represents the interquartile range, whiskers represent the maximum and minimum values, the middle indicates the median, and the x within the box represents the mean (A, *n* = 3; B, *n* = 6).

**Figure 3 plants-11-00606-f003:**
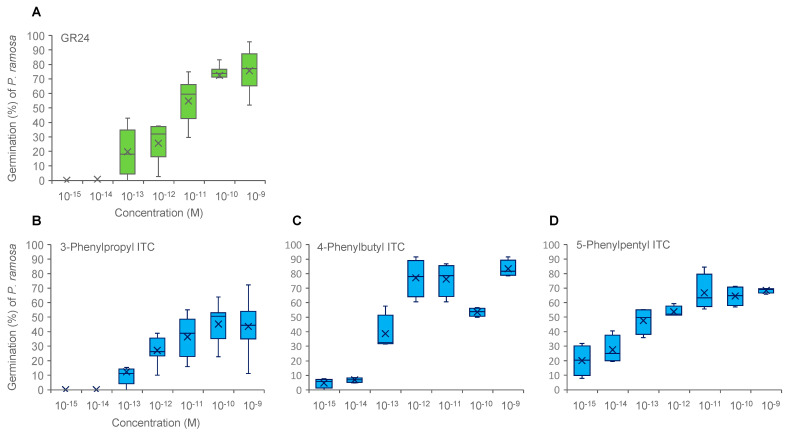
Germination stimulation activities of GR24 (**A**), 3-phenylpropyl ITC (**B**), 4-phenylbutyl ITC (**C**), and 5-phenylpentyl ITC (**D**) on *P. ramosa* at lower concentrations. The box represents the interquartile range, whiskers represent the maximum and minimum values, the middle indicates the median, and the x within the box represents the mean (*n* = 6).

**Figure 4 plants-11-00606-f004:**
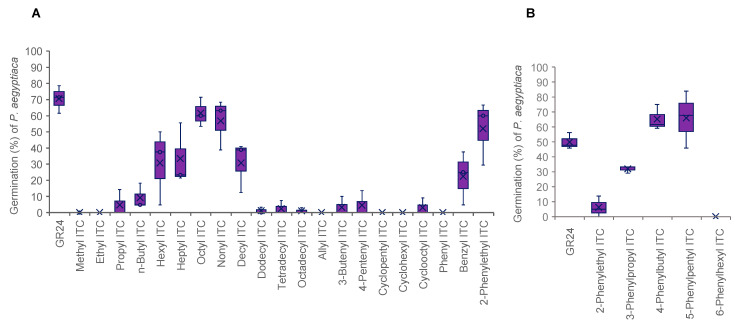
Germination stimulation activities of ITCs on *P. aegyptiaca* seeds. The activities were examined at 10^−6^ M (**A**) and 10^−8^ M (**B**). The box represents the interquartile range, whiskers represent the maximum and minimum values, the middle indicates the median, and the x within the box represents the mean (*n* = 3).

**Figure 5 plants-11-00606-f005:**
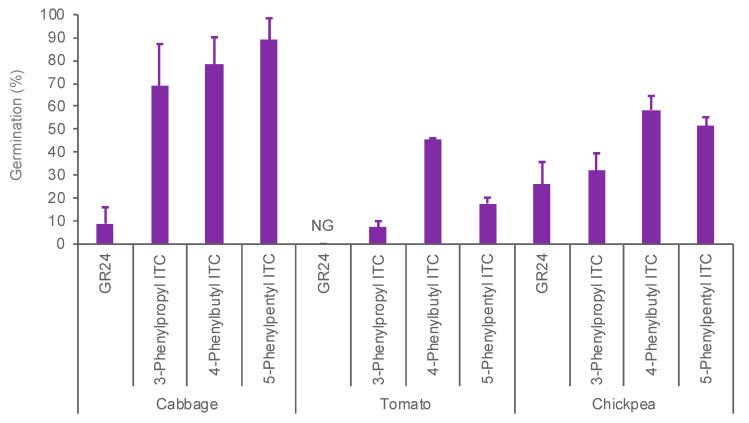
Effects of GR24 and ITCs on germination of *P. aegyptiaca* parasitizing different host plants including cabbage, tomato, and chickpea. Stimulant activities were examined at 10^−9^ M. NG means no germination. Bar means standard errors (*n* = 6).

## Data Availability

Data is contained within the article and Appendix A.

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
