# Peer review of "Germination Stimulant Activity of Isothiocyanates on Phelipanche spp."

_plants, 2022, doi:10.3390/plants11050606_

Round 1
Reviewer 1 Report
The present manuscript describes the testing of available 21 ITCs for Phelipanche spp. seed germination stimulation and some important structural features of ITCs. Although the idea is good, but the experimental setup is very simple thus the current study does not give information on how tested ITCs regulate Phelipanche spp. growth. My opinion it is a pilot experiment just good enough to select the suitable concentrations of TCs, more experiments of parasitic weed grown with host plants grown in soil to mimic the realistic conditions are importantly need
Author Response
Many thanks for your comments.
In this study, we would like to propose that ITCs may be good lead compounds for suicidal germination inducers. As in the case of SPL-7, a highly potent germination stimulant for Striga hermonthica, intrinsic activity of compound is of great importance. Therefore, in this study, we tried to obtain more active ITCs. In the next step, selected compounds will be tested their efficacy as suicidal germination inducers in the pot and field experiments in Israel as we are not allowed to do such experiments with Phelipanche seeds in Japan.
Reviewer 2 Report
The use of naturally produced chemicals is a fascinating concept with high potential to apply in the field. On the other hand, the use of synthetic compounds has a few associated concerns, such as their toxicity and biodegradability. How can we ensure that there are no residues? Will the residual compound interfere with the germination of the crop plants other than cabbage?
What could be the possible reasons for variation in sensitivity of chickpea and tomato to ITCs?
Please rectify the statement that plants release SLs to attract AM fungi.
Please note that plants release SLs to attract AM fungi and nitrogen-fixing bacteria as well. See Reference: Regulation of Plant Mineral Nutrition by Signal Molecules. Microorganisms. 2021, 9(4), 774; https://doi.org/10.3390/microorganisms9040774.
Author Response
The use of naturally produced chemicals is a fascinating concept with high potential to apply in the field. On the other hand, the use of synthetic compounds has a few associated concerns, such as their toxicity and biodegradability. How can we ensure that there are no residues? Will the residual compound interfere with the germination of the crop plants other than cabbage?
Many thanks for your comments. In general, ITCs decompose easily in soils and will not remain for long period. It was already reported that ITCs did not negatively affect tomato germination and growth in vitro as in the case of cabbage. Synthetic ITCs behave similarly as the major degradation is the hydrolysis of the isothiocyanate (NCS) group. Therefore, it is not likely that residual ITCs interfere with the germination of the crop plants. Of course, we would like to examine if ITCs cause any adverse effects on the germination and growth of crops in the soil in the future.
What could be the possible reasons for variation in sensitivity of chickpea and tomato to ITCs?
Thanks for your comments. It may be that they are slightly different biotypes (pathotypes) or there would be minor differences in their receptors for ITCs (and SLs). We are now analyzing the amino acid sequences of the receptor protein in these Phelipanche parasitizing cabbage, tomato and chickpea.
Please rectify the statement that plants release SLs to attract AM fungi.
Recent findings strongly suggest that SLs evolved as host recognition signals for AM fungi.
Please note that plants release SLs to attract AM fungi and nitrogen-fixing bacteria as well. See Reference: Regulation of Plant Mineral Nutrition by Signal Molecules. Microorganisms. 2021, 9(4), 774; https://doi.org/10.3390/microorganisms9040774.
Thanks for your comments. A few reports including this have been added.
Reviewer 3 Report
Parasitic plants of genera Orobanche and Phelipanche germinate after exposition to chemical signals exuded by roots of the host plants. The most studied germination stimulants belong to strigolactones, the newly discovered plant hormones which are stimulating hyphal branching of arbuscular mycorrhizal fungi and are involved in regulation of shoot and root architecture of plants.
The authors bring a unique discovery of new highly active stimulants of seeds of parasitic plants from genera Phelipanche without adversely affecting the germination and growth of the host plants (on example of cabbage), and that is the key finding and message of this study. The results point out also there are differences in the requirement for germination signals that possibly depend on the host of broomrapes and specific habitat.
I have some comments/notes
Abstract:
- add one or two sentences, at the beginning of the abstract, to provide a basic introduction to the field, comprehensible to a scientist in any discipline
- also, at the end of the abstract provide broader perspective (one sentence)
Introduction:
- Introduce more Phelipanche ramosa and Phelipanche aegyptiaca from worldwide agricultural point of view
- Clearly explain research hypothesis that was tested
Discussion:
- Explain what these statements mean: ...moderatly active ....highly active ...weak stimulant (line 171-172)
Methods:
- Introduce more in detail sites of seed collection in all countries, e.g., where in Israel? in one place? where in Kenya? (line 252-256)
- explain the reason for the different temperatures (line 272)
General:
- check English and the text for many typos (e.g., lines 22, 161, 174, 190, 220, 249 etc.)
- from time to time too much chemistry makes the text a bit confusing until the reader disappears
Author Response
Parasitic plants of genera Orobanche and Phelipanche germinate after exposition to chemical signals exuded by roots of the host plants. The most studied germination stimulants belong to strigolactones, the newly discovered plant hormones which are stimulating hyphal branching of arbuscular mycorrhizal fungi and are involved in regulation of shoot and root architecture of plants.
The authors bring a unique discovery of new highly active stimulants of seeds of parasitic plants from genera Phelipanche without adversely affecting the germination and growth of the host plants (on example of cabbage), and that is the key finding and message of this study. The results point out also there are differences in the requirement for germination signals that possibly depend on the host of broomrapes and specific habitat.
I have some comments/notes
Abstract:
- add one or two sentences, at the beginning of the abstract, to provide a basic introduction to the field, comprehensible to a scientist in any discipline
Many thanks for your comment. Revised as suggested.
- also, at the end of the abstract provide broader perspective (one sentence)
Many thanks for your comment. Revised as suggested.
Introduction:
- Introduce more Phelipanche ramosa and Phelipanche aegyptiaca from worldwide agricultural point of view
Many thanks for your comment. Revised as suggested.
- Clearly explain research hypothesis that was tested
Many thanks for your comment. Revised as suggested.
Discussion:
- Explain what these statements mean: ...moderatly active ....highly active ...weak stimulant (line 171-172)
Many thanks for your comment. Revised as suggested.
Methods:
- Introduce more in detail sites of seed collection in all countries, e.g., where in Israel? in one place? where in Kenya? (line 252-256)
Many thanks for your comment. Revised as suggested.
- explain the reason for the different temperatures (line 272)
Many thanks for your comment. Revised as suggested.
General:
- check English and the text for many typos (e.g., lines 22, 161, 174, 190, 220, 249 etc.)
Many thanks for your comment. Revised as suggested.
- from time to time too much chemistry makes the text a bit confusing until the reader disappears
Many thanks for your comment. Revised as suggested.
Round 2
Reviewer 1 Report
Author responses did not address my concerns. The experimental setup is very simple and does not improve our understanding on how tested ITCs regulate Phelipanche spp. growth. Germination is a very simple test, is only useful to screen for most active ITCs, then additional experiments/analyses on how active ITCs affect Phelipanche spp infection with host plants grown in soil to mimic the realistic conditions are importantly need.
This work is very limited and does not good enough to be published in high impact factor journal such as Plants
Author Response
We agree that it is important to understand how tested ITCs regulate growth of Phelipanche, in particular, under field conditions. In this study, as the reviewer 1 indicates, we just tried to select most active ITCs that may have potentials to be good leads for suicidal germination inducers of Phelipanche spp., because the intrinsic activity of a chemical is one of the important features determining its efficacy as for example pesticide.
We are sorry that we cannot conduct any pot or field experiments with Phelipanche spp. in Japan. We will test potentials of ITCs as suicidal germination inducers in the future, very likely in Israel. We are now proceeding the process.
Recently, several putative strigolactone receptors in P. ramosa have been identified and among them, PrKAI2d3 seems to be involved in seed germination stimulation (de Saint Germain et al. Plant Communications, 2021), although exact function of PrKAI2d3 remains elusive. Highly active ITCs may be useful chemical probes to understand the seed germination mechanisms of Phelipanche seeds.